# Model Predictive Control Strategy Based on Loss Equalization for Three-Level ANPC Inverters

**Shaoqi Wan [1], Bo Wang [2], Jingbo Chen [2], Haiying Dong [1,*] and Congxin Lv [1]**

[1] College of New Energy and Power Engineering, Lanzhou Jiaotong University, Lanzhou 730070, China; 12212033@stu.lzjtu.edu.cn (S.W.); 11210380@stu.lzjtu.edu.cn (C.L.)

[2] Lanzhou Wanli Airlines Electromechanical Limited Liability Company, Lanzhou 730070, China; wangb18306782.135@mail.avic (B.W.); chenjb18708803.135@mail.avic (J.C.)

\* Correspondence: hydong@mail.lzjtu.cn

**Abstract:** Targeting the issue of high losses of individual switching tubes in Neutral-Point Clamped (NPC) three-level inverters, an Active Neutral-Point Clamped (ANPC) three-level inverter is used, and a model predictive control strategy using the loss equalization of the inverter is proposed. This method organizes and analyzes multiple zero-state current pathway commutation modes and adds mode three under the original two commonly used zero-state commutation modes. On this basis, the three modes are flexibly switched by model predictive control, and the output is optimized according to the value function for the space vector in each operation, while the midpoint voltage control is added to the value function. The simulation results suggest that the recommended strategy in this study may effectively realize the loss equalization control and midpoint voltage control of the ANPC inverter, which improves the operation efficiency of the electromechanical actuator.

**Keywords:** three-level inverter; active midpoint clamp; loss equalization control; commutation mode; model predictive control; midpoint voltage



## 1. Introduction

Electromechanical servo systems are increasingly used in aerospace [1,2], and electromechanical actuators are the actuators of electromechanical servo systems, which are more efficient, more integrated, and easier to maintain than traditional hydraulic actuators. With the development of aerospace electromechanical actuators, permanent magnet synchronous motors (PMSM) have become an important part of them because of their high efficiency, high power density, and precise control [3,4]. The control effect of PMSM plays a crucial role in the performance and reliability of electromechanical actuator systems [5–7], and most of the existing researches improve the performance of PMSM by adopting multi-level inverters with advanced control strategies and the loss control should be considered by adopting multilevel inverters.

Among several multilevel structures, the NPC inverter circuit is currently one of the more mature structures, and its circuit structure and control method are straightforward and have the benefits of a high voltage withstand level and a low rate of output voltage distortion [8,9]. However, due to its loss, the problem is difficult to solve, resulting in some switching tubes' junction temperatures being too high and reducing the system efficiency, so the redundant zero-level path more ANPC inverter is used to solve the problem.

In the existing problem of ANPC inverter loss, soft-switching technology is used, which can effectively reduce the voltage and current stresses, switching losses, and electromagnetic interference of switching devices [10,11]. However, multiple switching tubes, as well as capacitors and inductors, need to be added to the inverter to generate resonance, which increases the device size and reduces the power density. Due to the high requirement of device power density in electromechanical actuators [1,2], it does not meet the requirements. Part of it combines the zero-level characteristics of the ANPC inverter,

replacing the high-frequency part of the switching tube Si(Silicon) IGBT with SiC(Silicon Carbide) MOSFETs to reduce the loss by using the characteristics of SiC devices [12–14]. This method reduces the loss, but the SiC device increases the cost, and the $\mathrm{d}i/\mathrm{d}t$ and $\mathrm{d}v/\mathrm{d}t$ of the two devices with different switching speeds in the switching process will lead to electromagnetic interference and switching overvoltage [12], so it does not meet the requirements; The remaining part of the improvement of the control strategy, through the reasonable allocation of the redundant zero-level path to achieve the balanced control of the switching tube loss.

Literature [15] based on hardware PWM configuration outputs asymmetric driving waveforms, which makes the turn-on and turn-off losses separate, but its calculation is complicated and does not make full use of the redundant zero level of the ANPC inverter. Literature [16] controls the ANPC inverter two redundant zero level allocation mode one and mode two alternately; for the different allocation modes to bear the switching loss device, different characteristics of the two flexible switching are needed to achieve the switching tube loss equalization control. Literature [17] combines the two methods to turn a single zero-level clamp circuit into two clamp circuits conducting at the same time, which reduces the conduction loss and increases the efficiency. Literature [18] regulates the frequency cycle, changing the duration ratio of mode one and mode 2 in each frequency cycle to realize the balanced adjustment of switching tube losses, but the ratio selection lacks a theoretical basis, and frequent switching between different modes will produce excessive switching losses, affecting the system efficiency [19]. Literature [16,20,21] used a model prediction method of multi-objective optimization for loss equalization control by setting the value function loss minimization but did not classify and make full use of the ANPC inverter redundancy zero level in a manner, whereas literature [20,21] used only manner two and its variants.

Based on the above problems, this paper makes complete use of the redundant zero level of the ANPC inverter and adds mode three on the basis of the two commonly used zero-level path modes. It summarizes the switching tubes and the loss law driven by each mode, designs the loss principle between the modes, and uses the model predictive control to realize the flexible switching on the basis of which the model predictive control is implemented [7,22]. Thus, the switching tube loss equalization control is accomplished, and the solution to the problem of uneven inverter loss in electromechanical actuators is realized, and, at the same time, it realizes inverter midpoint voltage equalization control. Finally, relevant simulations are carried out in Matlab to verify the theoretical analysis's correctness, as well as the feasibility and effectiveness of the recommended plan.

## 2. ANPC Inverter with Its Zero-Level Commutation Method

### 2.1. ANPC Inverter with Space Voltage Vectors

The three-level ANPC inverter topology is shown in Figure 1. In the figure, $C_1$ and $C_2$ are the upper and lower dc-side capacitors, respectively, and point o is the midpoint of the dc-side bus. $S_{a1}$, $S_{a2}$, $S_{a3}$, and $S_{a4}$ are the four IGBT switching tubes on the bridge arms of each phase, and the clamp diode in the original NPC inverter is replaced by $S_{a5}$ and $S_{a6}$, and there are antiparallel diodes $D_{a1}$–$D_{a6}$ connected to each switching tube. $U_{dc}$ is the dc-side bus voltage; io is the current, and $i_{c1}$ and $i_{c2}$ are the currents that flow through the dc-side capacitors $C_1$ and $C_2$. io is the midpoint current of the dc side, and $i_{c1}$ and $i_{c2}$ are the currents that flow through the dc side capacitors $C_1$ and $C_2$.

Same as the NPC inverter, each phase of the bridge arm has three modes, p, o, and n, which represent the three voltage amplitudes $U_{dc}/2$, 0, and $-U_{dc}/2$ of the output voltage $U_{dc}$. There are a total of $3^3 = 27$ switching states, which correspond to the space voltage vector, as shown in Figure 2.

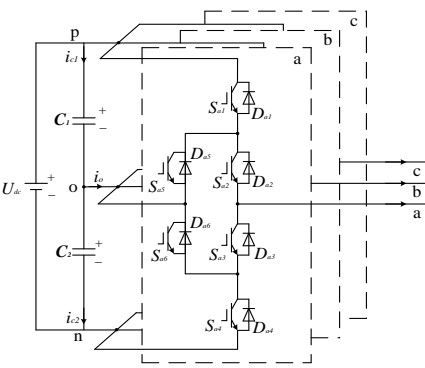

**Figure 1.** Typical circuit of the three-shunt sensing inverter.

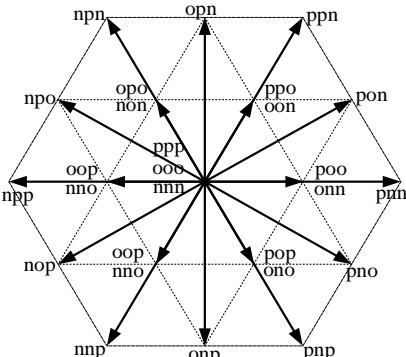

**Figure 2.** Space vector diagram.

Table 1 shows the space vectors corresponding to the vector types, where the small vectors appear in pairs and have an opposite effect on the midpoint voltage, the medium vectors cause the midpoint voltage to increase, while the zero and large vectors have no impact on the midpoint voltage.

**Table 1.** Table of vector types.

| | Switching State | Midpoint Current |
|---|---|---|
| Zero vector | ppp, ooo, nnn | 0 |
| Small vector | poo, onn<br>ppo, oon<br>opo, non<br>opp, noo<br>oop, nno<br>pop, ono | $-i_a, i_a$<br>$i_c, -i_c$<br>$-i_b, i_b$<br>$i_a, -i_a$<br>$-i_c, i_c$<br>$i_b, -i_b$ |
| Medium vector | opn, onp<br>pon, nop<br>pno, npo | $i_a$<br>$i_b$<br>$i_c$ |
| Large vector | pnn, ppn, npn<br>npp, nnp, pnp | 0 |

### 2.2. Space Voltage Vector Modeling

Assuming that the power electronic devices in the inverter are all ideal devices and equating the single-phase bridge arm to a single-pole, three-throw switch, with the three modes p, o, and n corresponding to the three switching states of each bridge arm, the switching function $S_x$ can be expressed in the following way:

$$S_x(x \in a, b, c) = \begin{cases} 1, & \text{p-level} \\ 0, & \text{o-level} \\ -1, & \text{n-level} \end{cases} \tag{1}$$

The phase voltage at the output of the three-level inverter can be expressed as:

$$\begin{cases} u_a = \frac{S_a}{2} U_{dc} \\ u_b = \frac{S_b}{2} U_{dc} \\ u_c = \frac{S_c}{2} U_{dc} \end{cases} \tag{2}$$

where $u_a$, $u_b$, $u_c$ are the phase voltages of the three phases of ABC, respectively; $U_{dc}$ is the DC bus voltage; $S_a$, $S_b$, and $S_c$ are the switching states of the three-phase bridge arms of ABC.

After the relationship between the line voltage as well as the three-phase current and the three-phase voltage at the space angle of $120°$ to each other, the three-phase output voltage synthesized vector is expressed as follows:

$$U = \frac{U_{dc}}{6} \times \begin{bmatrix} 2 & \frac{1}{2} - j\frac{\sqrt{3}}{2} & \frac{1}{2} + j\frac{\sqrt{3}}{2} \\ -1 & -1 + j\sqrt{3} & \frac{1}{2} + j\frac{\sqrt{3}}{2} \\ -1 & \frac{1}{2} - j\frac{\sqrt{3}}{2} & -1 - j\sqrt{3} \end{bmatrix} \times \begin{bmatrix} S_a \\ S_b \\ S_c \end{bmatrix} \tag{3}$$

The equivalent mathematical model of all 27 space voltage vectors can be obtained from the above equation.

### 2.3. ANPC Inverter Zero-Level Commutation Method and Corresponding Losses

The current paths in the p-level and n-level of the ANPC inverter are the same as in the NPC topology; while the o-level has more options, as shown in Table 2, the current in the o-level can circulate through $S_2$ and $S_5$ or through $S_3$ and $S_6$. Therefore, switching, depending on the switching tube turned on, can be categorized into six forms. Where the switching tube $S_{a1}$ with its anti-parallel diode $D_{a1}$ becomes $S_1$, and the other switching tubes are the same.

**Table 2.** Switch status table.

|      | $S_1$ | $S_2$ | $S_3$ | $S_4$ | $S_5$ | $S_6$ |
|------|-------|-------|-------|-------|-------|-------|
| p    | 1     | 1     | 0     | 0     | 1     | 1     |
| n    | 0     | 0     | 1     | 1     | 1     | 0     |
| OUL1 | 0     | 1     | 0     | 0     | 1     | 1     |
| OUL2 | 0     | 1     | 1     | 0     | 0     | 1     |
| OUL3 | 0     | 1     | 1     | 0     | 1     | 0     |
| OUL4 | 0     | 0     | 1     | 0     | 1     | 1     |
| OL1  | 1     | 0     | 1     | 0     | 0     | 1     |
| OL2  | 0     | 1     | 0     | 1     | 1     | 0     |

Since the bridge arm voltage will switch back and forth between p, o and n states during the inversion process, and because each switching process requires different switching tubes to be controlled, the six forms are categorized into three modes, mode one, mode two and mode three.

As demonstrated in Figure 3, mode 1 combines the OUL1 and OUL4 zero level mode, in the p-level switching o-level process, only $S_1$ and $S_5$ switch adjust, $S_2$ and $S_6$ switch states remain unchanged; similarly, from the n-level switching to the o-level process, only $S_4$ and $S_6$ switch to adjust, $S_3$ and $S_5$ switch state remains unchanged. (The arrows in the figure point to the switching tube that needs to change the state, where the dotted line indicates the switching tube that needs to be turned off and the other that needs to be turned on).

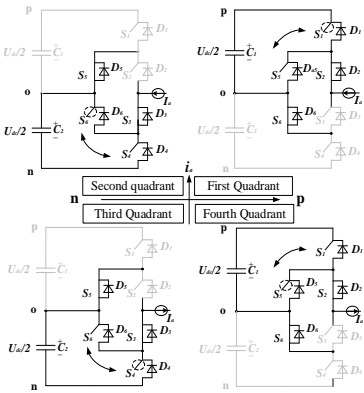

**Figure 3.** Zero-level path switching mode 1.

Therefore, in current path mode 1, the switching losses are concentrated in the $S_1$, $S_4$, $S_5$, and $S_6$ tubes, which reduces the losses in the $S_2$ and $S_3$ tubes, as shown in Figure 4 for the switching tube losses under mode 1 in the simulation, with higher overall losses $S_1$ and $S_4$. The loss collection background is: The test background is exactly the same as that in Section 5, which is the loss generated by the PMSM under model predictive control when running at steady state for 0.1 s at 3000 rpm and 5 N-m load conditions. Then the sampling frequency is set to 100 kHz, and the IGBT adopts Infineon's IKZ75N65EH5, whose turn-on and turn-off losses are 0.68 mj and 0.43 mj, respectively, at 25 °C; the initial voltage drop $V_{ceo}$ of the device is checked by the switching tube datasheet, which is 1.65 V, and the internal resistance r is 10 mΩ in the on-state.

**Figure 4.** Mode 1 switch tube loss.

As demonstrated in Figure 5, the mode 2 current path method combines the OUL2 and OUL3 o-level methods, in the process of p-level switching o-level only $S_2$ and $S_3$ switches are adjusted, and the $S_1$ and $S_6$ switch status remains unchanged; similarly, the process of switching from the n-level to the o-level is only adjusted for the $S_2$ and $S_3$ switches, and the $S_4$ and $S_5$ switches status remains unchanged.

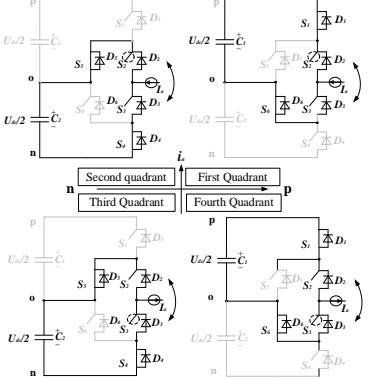

**Figure 5.** Zero-level path switching mode 2.

Therefore, in the current path switching mode of mode 2, the switching losses are concentrated in the $S_2$ and $S_3$ tubes, which reduces the losses of the rest of the switching tubes, as shown in Figure 6 for the switching tube losses under mode 2 in the simulation, and the overall losses $S_2$ and $S_3$ are higher.

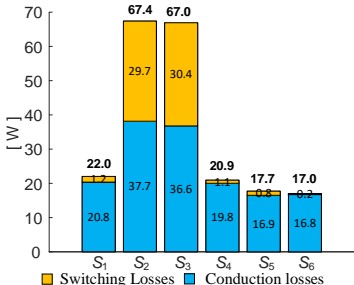

**Figure 6.** Mode 2 switch tube loss.

As shown in Figure 7 for mode three current path switching mode, which combines the OL1 and OU1 o-level mode, in the process of p-level switching o-level only $S_1$ and $S_3$ switch to adjust, $S_2$ and $S_6$ switch state remain unchanged; similarly, from the n-level switching to the o-level process is only adjusted to the $S_2$ and $S_4$ switch, $S_3$ and $S_5$ switch state remains unchanged.

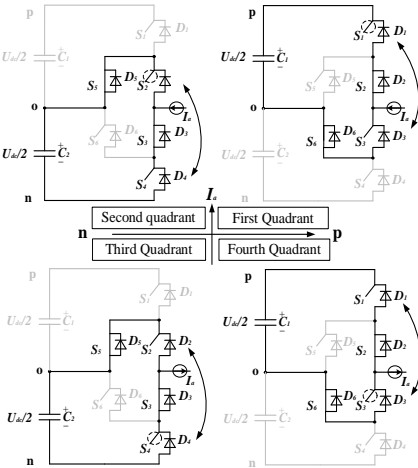

**Figure 7.** Zero-level path switching mode 3.

Therefore, in the current path switching mode of mode 3, the switching loss is concentrated in the $S_1$, $S_2$, $S_3$ and $S_4$ tubes, which reduces the loss of the $S_5$ and $S_6$ tubes, as shown in Figure 8 for the switching loss under mode 3 in the simulation, the loss of the $S_1$–$S_4$ tubes is more average, and the loss of the $S_5$ and $S_6$ tubes is lower compared to the previous two modes.

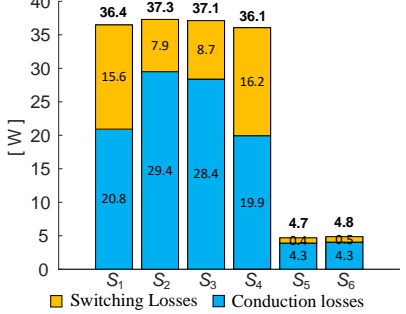

**Figure 8.** Mode 3 switch tube loss.

## 3. Switching Tube Loss Evaluation

Most of the inverter power switching tube are used with anti-parallel diode IGBT device composition, it is generally used fast recovery diode, because of its negligible turn-on loss and turn-off loss is much smaller than the IGBT, this paper focuses on the discussion of IGBT loss.

Losses are mainly switching losses and conduction losses of two types, of which switching losses include turn-on losses and turn-off losses.

### 3.1. ANPC Inverter Switching Loss Analysis

The device during the turn-on and turn-off transient loss is referred to as switching loss, the size of the device is mostly determined by the size and direction of the voltage and current on both sides of the device, and the effect is usually a nonlinear relationship, the representation is generally approximated by constructing a linear relationship.

Assume that the IGBT's switching frequency is $f_s$, in a switching frequency of switching change process in the role of time $[t_1, t_2]$, then the average switching loss in the role of time is roughly equal to:

$$P_{sw} = \int_{t_1}^{t_2} (E_{on} + E_{off}) \times f_S \times i(t) \, dt \tag{4}$$

where $E_{on}$ and $E_{off}$ are the turn-on loss energy and turn-off loss energy of the IGBT under the actual peak operating current, respectively (which can be found in the switching tube datasheet), $f_s$ is the switching frequency, and $i(t)$ is the real-time current.

### 3.2. ANPC Inverter Conduction Loss Analysis

Conduction loss is the loss generated by the IGBT during the conduction period. The size of the loss is determined by the length of its own conduction time and the size of the current flowing during conduction during an action time, and its expression can be approximated as:

$$P_{on} = V_{ceo} \times i(t) + i(t)^2 \times r \tag{5}$$

where $r$ is the on-state internal resistance, $V_{ceo}$ is the initial voltage drop of the device (which can be found in the switching tube datasheet); $i(t)$ is the actual on-state current.

Therefore, let the switching frequency of the IGBT be $f_s$, and the action time during the switching process at one switching frequency be $[t_1, t_2]$, then the average conduction loss during the action time can be expressed as:

$$P_{on\_T} = \int_{t_1}^{t_2} (V_{ceo} + i(t)r) \times i(t) dt \tag{6}$$

## 4. Model Predictive Control Loss Equalization Method

The basic principle of the model prediction algorithm is to forecast the response of the future era based on the current state, take the first item of the control sequence as the control quantity at the next moment, and then make the predicted response close to the set target through rolling optimization and feedback correction. This is accomplished by bringing the relevant variables into the value function and then comparing the variables with the smallest error in the finite set so that the control value gradually approaches the reference value.

### 4.1. PMSM Current Prediction Model

The model predictive control performance is heavily dependent on the mathematical model of the controlled object, and a mismatch of the controlled object parameters or other unmodeled dynamics can affect the performance [23]. In this section, an electromechanical actuator in normal operation is targeted to be modeled under the following assumptions: the influence of external factors such as temperature and frequency on the motor parameters

is not taken into account, space harmonics and the higher harmonic components of the magnetic field are neglected, rotor losses are not taken into account, and the stator windings are perfectly uniformly distributed in space. The final continuous domain mathematical model of the permanent magnet synchronous motor in the synchronous rotating d-q coordinate system can be expressed as:

$$
\begin{cases}
u_{\mathrm{d}} = Ri_{\mathrm{d}} + \frac{\mathrm{d}}{\mathrm{d}t}(L_{\mathrm{d}}i_{\mathrm{d}} + \psi_{\mathrm{f}}) - \omega_1(L_{\mathrm{q}}i_{\mathrm{q}}) \\
u_{\mathrm{q}} = Ri_{\mathrm{q}} + \frac{\mathrm{d}}{\mathrm{d}t}(L_{\mathrm{q}}i_{\mathrm{q}}) - \omega_1(L_{\mathrm{d}}i_{\mathrm{d}} + \psi_{\mathrm{f}})
\end{cases}
\tag{7}
$$

where $u_{\mathrm{d}}$, $u_{\mathrm{q}}$ are the d-axis and q-axis output voltages, $i_{\mathrm{d}}$ and $i_{\mathrm{q}}$ are the d-axis and q-axis output currents, $\psi_{\mathrm{f}}$ is the permanent magnet chain, $R$ is the stator resistance, and $\omega_1$ is the electrical angular frequency.

The output currents $i_{\mathrm{d}}$, $i_{\mathrm{q}}$ can be approximated using the forward Euler discretization method as:

$$
\begin{cases}
\frac{\mathrm{d}i_{\mathrm{d}}}{\mathrm{d}t} \approx \frac{i_{\mathrm{d}}(k+1) - i_{\mathrm{d}}(k)}{T_{\mathrm{c}}} \\
\frac{\mathrm{d}i_{\mathrm{q}}}{\mathrm{d}t} \approx \frac{i_{\mathrm{q}}(k+1) - i_{\mathrm{q}}(k)}{T_{\mathrm{c}}}
\end{cases}
\tag{8}
$$

where $i_{\mathrm{d}}(k + 1)$ and $i_{\mathrm{q}}(k + 1)$ are the d-axis and q-axis output currents at the time of $k + 1$, $i_{\mathrm{d}}(k)$, $i_{\mathrm{q}}(k)$ are the d-axis and q-axis output currents at the time of $k$, respectively, and $T_{\mathrm{c}}$ signifies the output current's control period.

Substitute (8) into (7), it is possible to obtain the compilation:

$$
\begin{cases}
i_{\mathrm{d}}(k+1) = (1 - \frac{RT_{\mathrm{c}}}{L_{\mathrm{d}}})i_{\mathrm{d}}(k) + \frac{\omega_1 L_{\mathrm{q}} T_{\mathrm{c}}}{L_{\mathrm{d}}}i_{\mathrm{q}}(k) + \frac{T_{\mathrm{c}}}{L_{\mathrm{d}}}u_{\mathrm{d}}(k) \\
i_{\mathrm{q}}(k+1) = -\frac{\omega_1 L_{\mathrm{d}} T_{\mathrm{c}}}{L_{\mathrm{q}}}i_{\mathrm{d}}(k) + (1 - \frac{RT_{\mathrm{c}}}{L_{\mathrm{q}}})i_{\mathrm{q}}(k) + \frac{T_{\mathrm{c}}}{L_{\mathrm{q}}}u_{\mathrm{q}}(k) - \frac{\omega_1 \psi_{\mathrm{f}} T_{\mathrm{c}}}{L_{\mathrm{q}}}
\end{cases}
\tag{9}
$$

where $u_{\mathrm{d}}(k)$ and $u_{\mathrm{q}}(k)$ are the components of the space voltage vector on the d-axis and q-axis at time of k, respectively. Through (9), The projected value of output current at $k + 1$ moments under the operation of any space voltage vector can be calculated using $k$ moments.

### 4.2. Current Control Value Function

After collecting the current $i_{\mathrm{d}}(k)$ and $i_{\mathrm{q}}(k)$ values, the mathematical model synthesized with the 27 voltage vectors is sequentially substituted into the prediction model. The expected values $i_{\mathrm{d}}(k + 1)$, $i_{\mathrm{q}}(k + 1)$ for the following moment of the 27 voltage vectors can then be derived, and then the value function is set to compare and find the optimization with the reference currents $i_{\mathrm{d}}^*(k + 1)$, $i_{\mathrm{q}}^*(k + 1)$. Specifically, it is to find the voltage vector whose predicted value is closest to the reference value, so the predicted values of all the voltage vectors differ from the reference value and squared. Then, optimization is done to find the optimal voltage vector. The value function expression is as follows:

$$
\Delta i_{\mathrm{dq}} = (i_{\mathrm{d}}(k+1) - i_{\mathrm{d}} * (k+1))^2 + (i_{\mathrm{q}}(k+1) - i_{\mathrm{q}} * (k+1))^2
\tag{10}
$$

### 4.3. Midpoint Voltage Control Value Function

The expression for the midpoint current of the three-level ANPC inverter at moment $k$ is:

$$
i_{\mathrm{np}}(k) = -(S_{\mathrm{a}}(k)i_{\mathrm{a}}(k) + S_{\mathrm{b}}(k)i_{\mathrm{b}}(k) + S_{\mathrm{c}}(k)i_{\mathrm{c}}(k))
\tag{11}
$$

where

$$
S_x = \begin{cases}
1, & \text{The } x^{\mathrm{th}} \text{ phase output p} \\
0, & \text{The } x^{\mathrm{th}} \text{ phase output o} \quad x = a, b, c \\
-1, & \text{The } x^{\mathrm{th}} \text{ phase output n}
\end{cases}
\tag{12}
$$

The expression for the upper and lower capacitance currents and capacitance voltages on the dc side at a time t is given by:

$$\begin{cases} i_{C1}(t) = C_1 \frac{dU_{C1}(t)}{dt} \\ i_{C2}(t) = C_2 \frac{dU_{C1}(t)}{dt} \end{cases} \tag{13}$$

Discretizing (13) yields:

$$\begin{cases} U_{C1}(k+1) = \frac{1}{C}i_{C1}(k)T_s + U_{C1}(k) \\ U_{C2}(k+1) = \frac{1}{C}i_{C2}(k)T_s + U_{C2}(k) \end{cases} \tag{14}$$

The offset of the upper and lower capacitor voltages on the DC side at $k+1$ is then:

$$\Delta U_{np}(k+1) = U_{C1}(k+1) - U_{C2}(k+1) = \frac{1}{C}i_{np}(k)T_s + \Delta U_{np}(k) \tag{15}$$

In the specific calculation process, this paper to the fastest inverter can change the switching frequency time as the role of time, for example, this paper to 100 kHz as the control algorithm sampling time, so $[t_1, t_2]$ is actually $[0, 1 \times 10^{-5}]$, that is, in every $1 \times 10^{-5}$ s to the loss generated in the previous moment of time for a calculation.

### 4.4. Loss Equalization Control Logic

The loss equalization control logic is shown in Figure 9. Firstly, the current value passed in each IGBT of the inverter is collected and entered into the loss model to calculate the current loss value, and the cumulative loss of each switching tube is calculated.

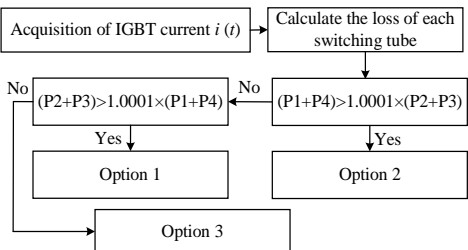

**Figure 9.** Loss equalization control logic block diagram.

Then, the loss judgment is performed, and the loss judgment time is based on multiple verifications to finally take the value of the electrical frequency, which is carried out once every 0.005 s, with the goal of equalizing the loss of $S_1$–$S_4$ tubes and making $S_5$ and $S_6$ as small as possible. If the sum of the losses of $S_1$ and $S_4$ tubes is greater than the sum of the losses of $S_2$ and $S_3$ tubes within a certain range, then the zero level is set to the mode two pathway; conversely, the zero level is set to the mode one pathway; if the losses of $S_1$ and $S_4$ tubes are approximately equal, then the zero level is set to the mode three pathway, and the judgment is set to judge the range as large as possible, to decrease the switching loss caused by toggling the three modes. After selecting the zero-level access path, the next judgment is made.

### 4.5. Finite Set Model Predictive Control

The finite set model prediction algorithm first performs rolling optimization on all voltage vectors in the finite set and then selects the voltage vector with the smallest value function for output. Each rolling is computed for the finite set's 27 voltage vectors.

Figure 10 depicts the total system block diagram, which includes model predictive control for the current inner loop and PI control for the speed outer loop. Firstly, the reference rotational speed $\omega^*$ is given, and then the difference with the collected actual rotational speed $\omega$ is made and brought into the PI regulator. Next, the reference value of stator q-axis current $i_q^*(k+1)$ is obtained by the PI regulator and brought into the

value function with the predicted value for calculation, and then the optimal vector is selected by searching for the optimal vector, and the corresponding switching sequence is substituted into the inverter to output the three-phase currents to be supplied to the PMSM, in which the model predictive control controls the currents and the mid-point voltage of the inverter at the same time, that is, (10) and (15) are added together to finally value function is obtained as follows:

$$T = \Delta i_{dq} + n \times \Delta U_{np}(k+1) \tag{16}$$

where $n$ is the midpoint voltage control weight coefficient, and the control of current and midpoint voltage balance is realized by adjusting the $n$ value, In this paper, the value of $n$ is selected as 3500.

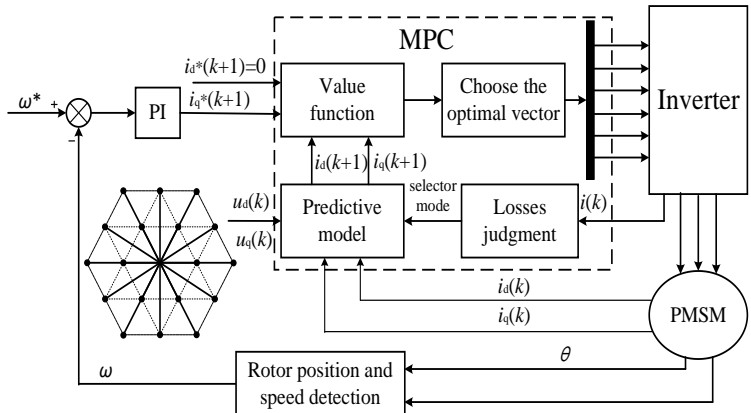

**Figure 10.** Model predictive control system block diagram.

Where $\omega$ and $\omega^*$ are the actual value and the given value of the motor speed, respectively, $i_d^*(k + 1)$, $i_q^*(k + 1)$ are the given values of stator d-axis and q-axis currents at the moment $k + 1$.

The predictive control technique is implemented as follows:

Step 1: The current $i(k)$, the inverter upper and lower capacitor voltages $U_{up}(k)$, $U_{down}(k)$ with the stator dq-axis currents $i_d(k)$, $i_q(k)$ in the PMSM flowing in each switching tube of the inverter at moment $k$ are collected;

Step 2: Bring the current $i(k)$ into the inverter loss model (4) and (6) for loss calculation, and select the zero-level path for the next moment by judging the switching tube loss.

Step 3: After selecting the mode, the collected stator dq-axis currents $i_d(k)$, $i_q(k)$ with the 27 voltage vector mathematical models are brought into the prediction model (9) to obtain the expected currents $i_d(k + 1)$, $i_q(k + 1)$ for each voltage vector's $k + 1$ instant;

Step 4: Substitute the predicted currents with the reference currents $i_d^*(k + 1)$, $i_q^*(k + 1)$, and the collected midpoint voltages into the value function (16), calculate and select the voltage vector corresponding to the minimum value of the value function as the optimal vector, and finally act the optimal vector corresponding to the switching sequence on the inverter.

## 5. Simulation Verification

To test the validity and efficacy of the algorithm suggested in this work, based on the model prediction loss equalization control, Matlab simulation was carried out to compare and analyze the ANPC inverter and permanent magnet synchronous motor. The three phase PMSM is controlled using $i_d = 0$. Table 3 displays the simulation parameters. All of the following loss control backgrounds are the same as in Section 2.3, with only improvements to the control algorithms.

**Table 3.** Simulation Parameter.

| Parameters | Value | Unit |
|---|---|---|
| Control frequency | 100 | [kHz] |
| DC side voltage | 270 | [V] |
| DC side capacitance | 4700 | [μF] |
| d-axis inductance | 0.395 | [mH] |
| q-axis inductance | 0.395 | [mH] |
| Stator resistance | 0.0485 | [Ω] |
| Permanent magnet flux | 0.1194 | [Wb] |
| Number of pole-pairs | 4 | [pair] |
| Maximum motor speed | 6000 | [r/min] |
| Locked-rotor torque | 33 | [N·m] |
| Rated power | 15 | [kW] |

Figure 11 shows the loss of each switching tube in phase A of the ANPC inverter when only mode one and mode two are used under model control with the parameters of Table 3; Figure 12 shows the loss of each switching tube in phase A of the inverter with the addition of mode 3. In contrast, the improved algorithm has a better control effect, and the overall loss at the same time is reduced by 5.6%, of which the loss of the clamped switch tubes $S_5$ and $S_6$ is reduced by 24.6%, which proves the superiority of the control effect under the improved algorithm.

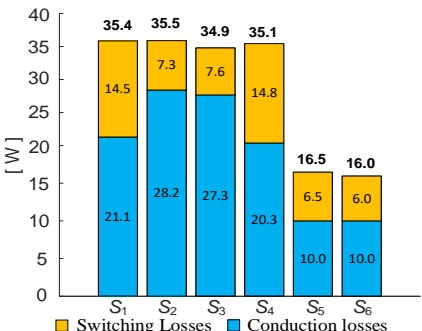

**Figure 11.** The traditional way of switching tube loss.

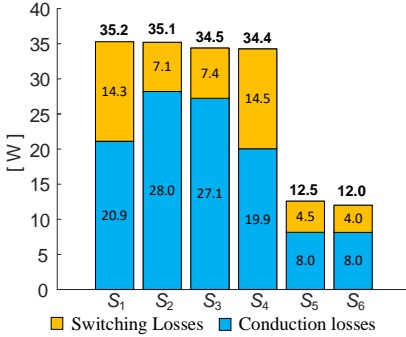

**Figure 12.** Improved switching tube loss.

Figures 13–18 give the waveforms of the motor operation at the motor setting of 5 N-m and 3000 rpm under the model predictive control with loss equalization. Under this condition, we collect and calculate the average switching frequency of the ANPC inverter, which is 18.7 kHz.

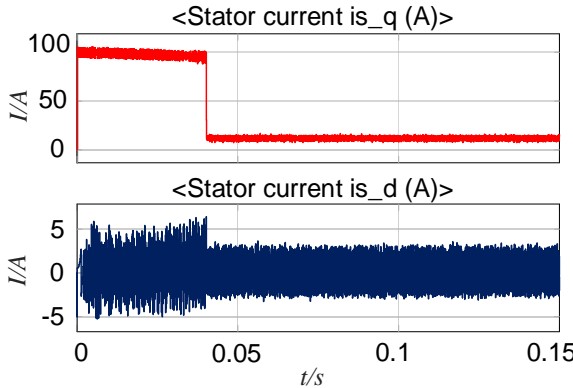

**Figure 13.** Motor stator dq axis current waveform.

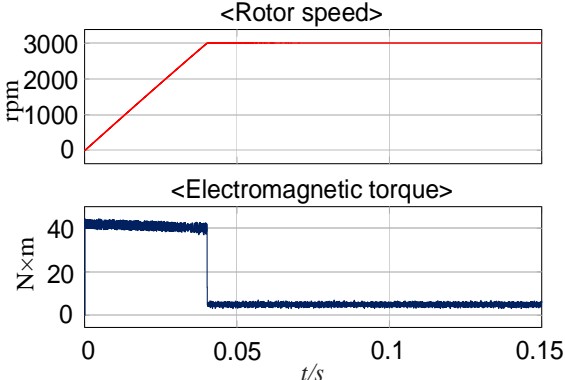

**Figure 14.** Motor speed and torque waveform.

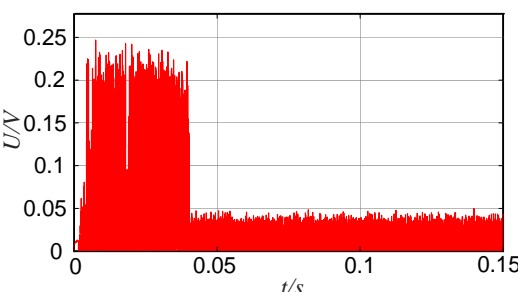

**Figure 15.** The waveform of the absolute value of midpoint voltage.

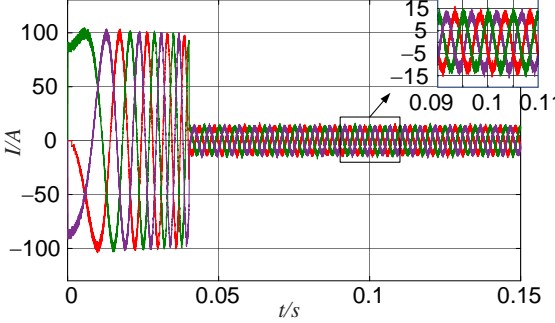

**Figure 16.** Three-phase current output waveform.

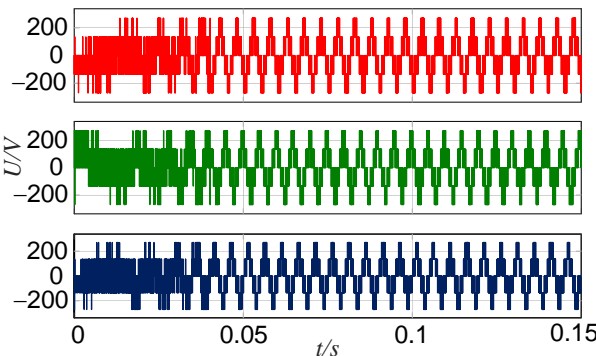

**Figure 17.** Three-phase voltage output waveform.

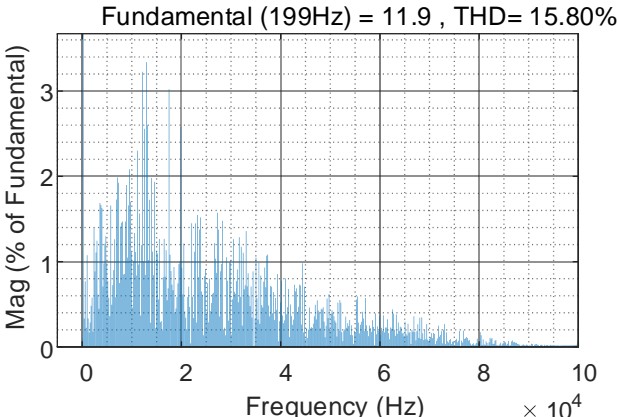

**Figure 18.** Current total harmonic distortion.

Where: In Figures 16 and 17, the red waveform represents phase a, the green waveform represents phase b, the blue waveform represents phase c.

From Figures 13 and 14, it can be seen that the motor reaches 3000 rpm in 0.04 s and starts to run steadily, at which time the motor stator dq-axis current and torque remain stable.

Figure 15 gives the midpoint voltage control effect of the ANPC inverter under the model predictive control of loss equalization, and it is clear that after the motor speed reaches 3000 rpm and stabilizes, the absolute value of the midpoint upper and lower capacitance voltages is kept within 0.05, which has a better control effect.

Figures 16 and 17 give the three-phase current and three-phase voltage output waveforms of the inverter under the predictive control of the model with loss equalization. It can be seen that the three-phase current and three-phase voltage remain stable after the motor speed reaches 3000 rpm.

Figure 18 shows the total harmonic distortion of phase A current in Figure 16 when the motor is stabilized at 3000 rpm. The three-phase current frequency is measured to be 199 Hz, and it can be seen that at a switching frequency of 100 kHz, the total harmonic distortion of the current is 15.8% and the current is slightly harmonic but generally stable.

Figures 19 and 20 give the waveforms of motor speed and torque during load up and load down, in which the motor speed is set to 3000 rpm, and the torque is switched between 0 N·m and 5 N·m. It can be seen that the rotational speed remains stable during load up and load down, the torque follows the target quickly and then remains stable, and there is no large torque pulsation in the waveforms.

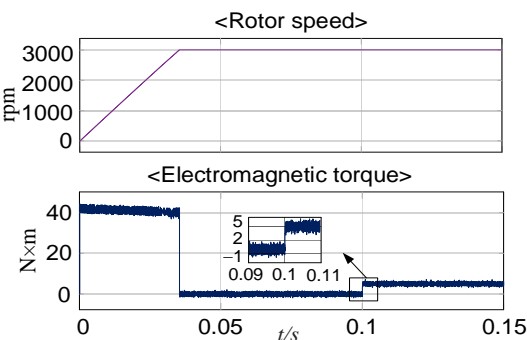

**Figure 19.** Speed and torque waveform in load lifting.

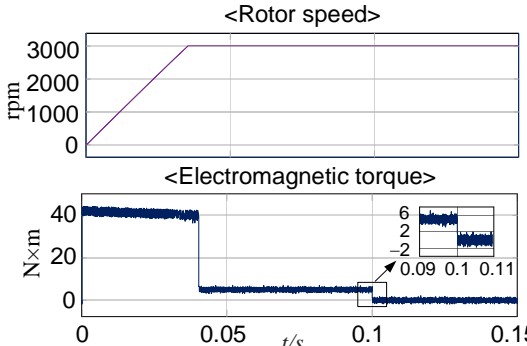

**Figure 20.** Speed and torque waveform in load shedding.

Figures 21 and 22 give the waveforms of motor speed and torque during acceleration and deceleration, in which the motor torque is set to 5 N·m, and the speed is switched between 1000 and 3000 rpm.

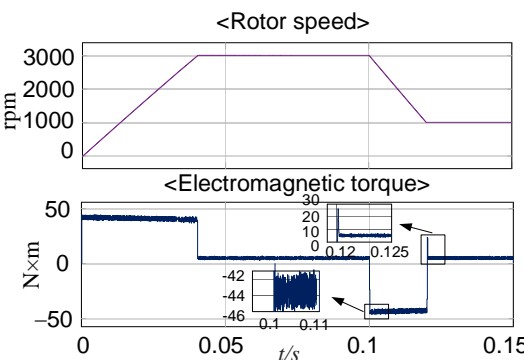

**Figure 21.** Speed and torque waveform in deceleration.

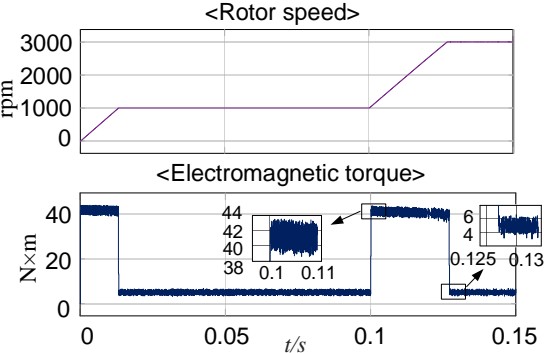

**Figure 22.** Speed and torque waveform in acceleration.

As can be seen in Figure 21, the motor can be decelerated from 3000 rpm to 1000 rpm in 0.02 s, the torque decreases rapidly and remains stable at the moment of deceleration, and the torque quickly follows and remains stable after the momentary torque pulsation when the deceleration is completed; as can be seen in Figure 19, the motor can be accelerated from 1000 rpm to 3000 rpm in 0.027 s, and the torque rapidly decreases and remains stable at the moment of acceleration, and the torque does not have large torque pulsation and quickly follows and remains stable at the moment of acceleration. When the acceleration is completed, the torque does not have a large torque pulsation, follows quickly, and remains stable.

Figure 23 gives the servo response waveform of the motor during the sudden change of load, in which the displacement of the motor-driven ball screw is set to change by 16 mm in 0.05 s, and the load is set to increase from 5 N·m to 8 N·m in 0.6 s. As shown in the figure, the red line represents the set reference position, and the green line represents the actual position. It can be seen from the figure that the ball screw reaches the preset position in 0.35 s, and at the same time, it remains stable and the position fluctuation is kept within ±0.05 mm in 0.4 s; After the sudden change of load in 0.6 s, the position of the ball screw remains basically unchanged, slightly shifted downward, but the position fluctuation is still kept within ±0.05 mm. mm; in 0.6 s after a sudden change in load, the position of the ball screw remains basically unchanged, slightly shifted downward, but the position fluctuation is still maintained within ±0.05 mm.

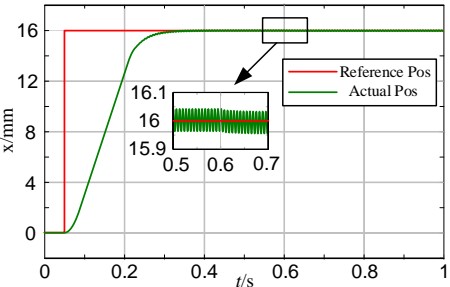

**Figure 23.** Load sudden change servo response waveform.

Figure 24 gives the servo response waveform of the motor in the displacement change process, in which the ball screw displacement change is set to 16 mm in 0 s, the displacement change is restored to 0 mm in 0.5 s, and the load is set to 5 N·m. From the figure, it can be seen that the ball screw position can be guaranteed to complete the servo response within 0.35 s in both the forward and backward movements and the position fluctuations are kept within ±0.05 mm.

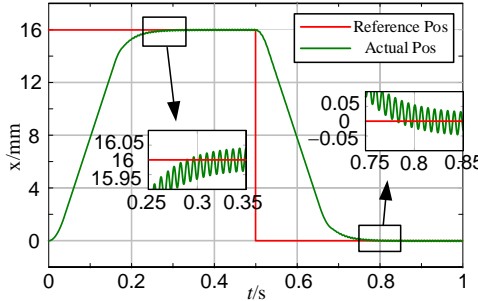

**Figure 24.** Displacement transform servo response waveform.

## 6. Conclusions

In this research, a model predictive control technique according to inverter loss equalization is presented to add pathway mode three under the original two commonly used zero-level pathway modes of ANPC inverters, and the three modes are flexibly switched

using model predictive control, and the following conclusions can be drawn through the theoretical analysis and simulation verification.

1.  The method in this research only improves the driving strategy to avoid the problem of power density reduction in electromechanical actuators and ensures the stability and rapidity of PMSM operation.
2.  Under the model predictive control, the improved algorithm realizes the ANPC inverter switching tube loss equalization, and at the same time reduces the overall loss of switching tubes, in which the loss of clamped switching tubes is greatly reduced.
3.  The method in this paper uses a model predictive control strategy to simultaneously achieve inverter switching tube loss equalization and midpoint voltage control to improve the operating efficiency of the electromechanical actuator.
4.  The method in this paper only changes the driving strategy, which reduces the overall inverter loss as well as the clamped switching tube loss. It can be generalized to the Si and SiC hybrid method for its characteristics.
5.  In this paper, the value of its time in the loss judgment is initially selected as 0.005 s each time, but whether this value is also applicable to other rotational speeds is not further verified in this paper, to be followed by further research.

**Author Contributions:** Conceptualization, S.W. and C.L.; methodology, S.W. and H.D.; software, S.W. and C.L.; validation, H.D. and C.L.; investigation; resources, B.W. and J.C.; data curation, S.W.; writing—original draft preparation, S.W.; writing—review and editing, H.D.; supervision, B.W. and H.D.; project administration; funding acquisition, B.W. and H.D. All authors have read and agreed to the published version of the manuscript.

**Funding:** This study was supported by the Major Science and Technology Program of Gansu Province, China (21ZD4GA005). Sponsor: Haiying Dong.

**Data Availability Statement:** The data that has been used are confidential.

**Acknowledgments:** The completion of this study is due to the collaborative efforts of several co-authors.

**Conflicts of Interest:** Author Shaoqi Wan was employed by College of New Energy and Power Engineering, Lanzhou Jiaotong University, The other authors declare no conflicts of interest.

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
