# Peer review of "Model Predictive Control Strategy Based on Loss Equalization for Three-Level ANPC Inverters"

_actuators, doi:10.3390/act13030111_

Round 1
Reviewer 1 Report
Comments and Suggestions for Authors
To support the main idea, the following comments should be addressed as
1) Loss Equalization: it should be clearly defined and explained for the main issue of the proposed manuscript.
2) Switch Modulation & Control: They should be addressed according to the proposed method.
3) Experimental Results: If possible, the authors are encouraged to show the experimental results to evaluate the proposed idea.
Comments on the Quality of English LanguageSome abbreviations should be modified as
1) NPC: Neutral-Point Clamped
2) ANPC: Active Neutral-Point Clamped.
In addition, Si and SiC should be clearly represented in Introduction as
1) Si: Silicon
2) SiC: Silicon Carbide.
Author Response
Dear Reviewers:
Thank you very much for your professional comments on our article. As you may have noticed, there are several issues that need to be addressed. Based on your good suggestions, we have made extensive changes to our previous draft. As a result, we have uploaded a copy of the original draft with all changes highlighted using the Track Changes mode in MS Word. Our point-by-point response to the comments is attached below. While reproducing the comments, our responses are directly highlighted in a different color (red).
- Loss Equalization: it should be clearly defined and explained for the main issue of the proposed manuscript.
We have added the main issues addressed in this paper in the last part of the introduction, while the main issues to be addressed have already been described in the beginning part of the abstract.
- Switch Modulation & Control: They should be addressed according to the proposed method.
In this paper, we optimize the space voltage vector as a finite set by model predictive control and output the switching sequence corresponding to the optimal space voltage vector into the inverter.
- Experimental Results: If possible, the authors are encouraged to show the experimental results to evaluate the proposed idea.
Due to the limitations of the study, it was not possible to experimentally validate the ideas proposed in this paper, and experimental validation is sought in subsequent studies.
4.Some abbreviations should be modified as
1) NPC: Neutral-Point Clamped
2) ANPC: Active Neutral-Point Clamped.
In addition, Si and SiC should be clearly represented in Introduction as
1) Si: Silicon
SiC: Silicon Carbide.
We have corrected the problem with the above abbreviations in the abstract and introduction.
We hope that the revised manuscript is accepted for publication in Actuators.
Sincerely yours,
Shaoqi Wan
Reviewer 2 Report
Comments and Suggestions for Authors
The paper presents the predictive control of an ANPC topology, where the goal is to balance the midpoint voltage and to equalize the switching loss across S1 and S4 transistors.
In the case of a scientific article, it is important to present the results in such a way that, based on the description, the results can be reproduced. As many parameters are not given, furthermore the description is not clear, it would be hard to reproduce the simulation results presented in the paper. Please improve the readability of the paper and add some more explanation!
My comments:
On page 11: "the on-state internal resistance, r, is 10Ω". Is it really 10 Ohm?
Please summarize the calculation requirements of the proposed method!
The sampling frequency is set to 100 kHz. What is the equivalent switching frequency and THD of the current at the investigated working point?
The loss judgment is carried out every 0.005s (page9, line 245), which means 200 Hz (MPC sampling frequency is 100 kHz). Why was this value selected?
The steps of the control algorithm are given in page 10. Step 2 is the loss calculation and the selection of the mode. Is it carried out every sampling instant? (see my previous comment, the loss judgement is carried out at 200 Hz, the MPC sampling frequency is 100 kHz).
Figure 14 present the time function of the current. Theoretically there is a stepwise change in the loading torque at 0.1 sec. However, no change can be seen in the time function of the phase current.
Please plot the time function id and iq currents to validate that the MPC works properly!
Please give the rated parameters of the PMSM (like rated current, voltage, speed, torque...)!
The machine is accelerated by 40 Nm. Why the loss is evaluated at 5 Nm, if the machine can handle higher torque as well (most likely the machine has higher rated torque)!?
There are same results (see Fig. 20 and 21), which presents waveforms obtained for a motor-driven ball screw system. Since no parameters are specified for the system, it is difficult to evaluate the obtained results.
What is the selected value of n weighting factor in the cost function given in (16)?
Please use the dot symbol to denote multiplication (the star symbol also used to represent reference value).
The prediction is carried out on 27 voltage vectors. There are redundant voltage vectors for a 3L inverter as it is shown in the paper. Is it necessary to calculate the prediction for all 27 vectors?
One well-known disadvantage of real-time digital implementation of MPC techniques is the required high processing capability due to the extensive number of calculations. It results in a delay between the measurements and the actuation, which can deteriorate the performance of the drive if it is not considered and not compensated. Have you implemented delay compensation in the simulation model?
Comments on the Quality of English LanguageModerate editing of English language required.
Author Response
Dear Reviewers:
Thank you very much for your professional comments on our article. As you may have noticed, there are several issues that need to be addressed. Based on your good suggestions, we have made extensive changes to our previous draft. As a result, we have uploaded a copy of the original draft with all changes highlighted using the Track Changes mode in MS Word. Our point-by-point response to the comments is attached below. While reproducing the comments, our responses are directly highlighted in a different color (red).
- On page 11: "the on-state internal resistance, r, is 10Ω". Is it really 10 Ohm?
Sorry, we checked and found that the unit of on-resistance is wrong, the correct unit is 10mΩ.
- Please summarize the calculation requirements of the proposed method!
The methodology proposed in this paper is applicable to the loss control of ANPC three-level inverters in aero-mechatronic actuators, where specific requirements and assumptions have been supplemented in section4.1.
- The sampling frequency is set to 100 kHz. What is the equivalent switching frequency and THD of the current at the investigated working point?
The total harmonic distortion of the phase A current at the operating point has been measured, i.e., as shown in Fig. 18.
- The loss judgment is carried out every 0.005s (page9, line 245), which means 200 Hz (MPC sampling frequency is 100 kHz). Why was this value selected?
Thank you for your question. Since each loss judgment means that the loss control method of the ANPC inverter needs to be changed (the zero level path needs to be changed), if the zero level path is changed too many times, it will also result in the accumulation of loss, and it is impossible to make an accurate judgment of the accumulation of loss within a short period of time due to the close proximity of the values. Therefore, for the time of loss judgment, this paper tested a number of times in the simulation verification process, and finally took 0.005 as the loss judgment time.
- The steps of the control algorithm are given in page 10. Step 2 is the loss calculation and the selection of the mode. Is it carried out every sampling instant? (see my previous comment, the loss judgement is carried out at 200 Hz, the MPC sampling frequency is 100 kHz).
Thank you for your question, for the loss calculation is done at each sampling instant and the accumulation operation is performed, while the mode selection time i.e. each loss judgment is performed, i.e. 0.005s.
- Figure 14 present the time function of the current. Theoretically there is a stepwise change in the loading torque at 0.1 sec. However, no change can be seen in the time function of the phase current.
Fig. 14 Fig. 15 shows the normal waveform when there is no sudden change in load and speed under loss equalization control, which I have not emphasized in the description thus giving you a misunderstanding and we have corrected it in the manuscript.
- Please plot the time function id and iq currents to validate that the MPC works properly!
Thank you for your question, at first we logged the data for the dq axis current and inadvertently did not put it in the article, it has now been added.
- Please give the rated parameters of the PMSM (like rated current, voltage, speed, torque...)!
Thanks to your comments, we have added the maximum motor speed, locked-rotor torque, and rated power in Table 3 of this paper.
- The machine is accelerated by 40 Nm. Why the loss is evaluated at 5 Nm, if the machine can handle higher torque as well (most likely the machine has higher rated torque)!?
After calculating the drive force for a ball screw with a rated speed of 3000 r/min and a transmission ratio of 20:1, it was finally concluded that only 5.37 N-m was required to satisfy the need, so only 5 Nm was evaluated.
- There are same results (see Fig. 20 and 21), which presents waveforms obtained for a motor-driven ball screw system. Since no parameters are specified for the system, it is difficult to evaluate the obtained results.
Similarly, Fig. 20 and Fig. 21 are evaluated based on the parameter of 5 Nm, and further research will be done next for other pressures that the control system can withstand.
- What is the selected value of n weighting factor in the cost function given in (16)?
As the value of the weighting factor is not proposed in the current more accurate method, so this paper in the experiment to test the selection of more excellent results, the n value in this paper is selected as 3500, and in the article section4.5 at the supplement.
- Please use the dot symbol to denote multiplication (the star symbol also used to represent reference value).
We have replaced the star symbol with the use of dot symbol for equation (16) in the text.
- The prediction is carried out on 27 voltage vectors. There are redundant voltage vectors for a 3L inverter as it is shown in the paper. Is it necessary to calculate the prediction for all 27 vectors?
The 3L inverter has redundant voltage vectors but each redundant vector produces an opposite effect on the midpoint voltage, which is mentioned in Table 1 of this paper.
- One well-known disadvantage of real-time digital implementation of MPC techniques is the required high processing capability due to the extensive number of calculations. It results in a delay between the measurements and the actuation, which can deteriorate the performance of the drive if it is not considered and not compensated. Have you implemented delay compensation in the simulation model?
Thank you for your question, we considered the issue and we added a delay module at the predicted output of the model with a delay of the sampling time 1e-6.
Finally, thank you again for your questions and comments, which have been very useful in improving the quality of my articles! And we hope that the revised manuscript is accepted for publication in Actuators.
Sincerely yours,
Shaoqi Wan
Reviewer 3 Report
Comments and Suggestions for Authors
The paper deals with a new predictive control based on loss equalization for 3-level ANPC inverters. The introduction is well written and with a sufficient number of references.
Section 2 explains the ANPC Inverter modulation strategies. The space voltage vector is a well kwown technique, however the explanation is clear.
In 2.3 the power losses has been analyzed. In fig. 4, fig. 6, fig. 8 is not clear the condition of the inverter (switching frequency?, kind of power devices?). Switching frequency and kind of pover device should modify che ratio between conduction losses and switching losses.
Section 4 deals with the control for the loss equalization. The PMSM model is clear, but the section 4.2 (current control value function) should be better explained, since in my opinion, is not fully clear.
Also Section 4.5 shoul be enriched with a few details.
Simulation results are clear and convincing.
Minor:
* Fig. 10 is barely legible.
* Table 3. khz -> kHz
English language if fine
Author Response
Dear Reviewers:
Thank you very much for your professional comments on our article. As you may have noticed, there are several issues that need to be addressed. Based on your good suggestions, we have made extensive changes to our previous draft. As a result, we have uploaded a copy of the original draft with all changes highlighted using the Track Changes mode in MS Word. Our point-by-point response to the comments is attached below. While reproducing the comments, our responses are directly highlighted in a different color (red).
- In 2.3 the power losses has been analyzed. In fig. 4, fig. 6, fig. 8 is not clear the condition of the inverter (switching frequency?, kind of power devices?). Switching frequency and kind of pover device should modify che ratio between conduction losses and switching losses.
We have added the inverter conditions (switching frequency and kind of power devices) in fig. 4, fig. 6, fig. 8 in the text.
- Section 4 deals with the control for the loss equalization. The PMSM model is clear, but the section 4.2 (current control value function) should be better explained, since in my opinion, is not fully clear.
We have enriched the introduction to section 4.2 (current control value function) to make it clearer.
- Also Section 4.5 shoul be enriched with a few details.
We have added to the details in Section 4.5, specifically to fig.10, and to the origin of equation (16).
4.Minor:
* Fig. 10 is barely legible.
We have adjusted the content distribution as well as the size of Fig.10.
* Table 3. khz -> kHz
we have corrected the “khz” into “kHz”.
We hope that the revised manuscript is accepted for publication in Actuators.
Sincerely yours,
Shaoqi Wan
Reviewer 4 Report
Comments and Suggestions for Authors
Reducing converter losses always remains an urgent task.
The authors proposed a full-fledged theoretical solution for predictive loss control in a three-level inverter.
A mathematical description of the forecasting model, control algorithm and loss assessment is given.
Current and correct links to literature sources.
The results are confirmed by oscillograms
The findings are complete and reflect the results of the study.
What environment did you model in? What conditions and assumptions were met during the modeling!
Increase the quality and size of Figure 13-19 with waveforms
Author Response
Dear Reviewers:
Thank you very much for your professional comments on our article. As you may have noticed, there are several issues that need to be addressed. Based on your good suggestions, we have made extensive changes to our previous draft. As a result, we have uploaded a copy of the original draft with all changes highlighted using the Track Changes mode in MS Word. Our point-by-point response to the comments is attached below. While reproducing the comments, our responses are directly highlighted in a different color (red).
- What environment did you model in? What conditions and assumptions were met during the modeling!
We have added the modeling environment as well as the assumptions at section 4.1.
- Increase the quality and size of Figure 13-19 with waveforms
We have improved the quality of Fig.13-19 by increasing the size to make the display clearer.
We hope that the revised manuscript is accepted for publication in Actuators.
Sincerely yours,
Shaoqi Wan
Round 2
Reviewer 2 Report
Comments and Suggestions for Authors
Authors answered all of my queries and they improved their initial submission by adding new explanation and new results as well.
I still have few concerns listed below:
- on page five line 142: "The inverter switching frequency is set to 100kHz" 100 Khz is the sampling frequency of the MPC control algorithm, but as Finite-Set MPC is utilized, most likely the switching frequency is much less than the sampling frequency. It can happen this results were made by a constant switching frequency (it is not clear from the text) without the proposed MPC scheme. In that case, the selected 100 kHz is very large for the IGBT taking into account for example the delays in the switching.
- page 15, line 366: "switching frequency of 100khz". As I indicated in my previous comment, 100 kHz is the sampling frequency of the control algorithm, and the switching frequency should be much lower. Please calculate the average number of switching and indicate in the paper. I suggest to use the method introduced in the paper (see equation 5):
P. Karamanakos and T. Geyer, “Guidelines for the design of finite control set model predictive controllers,” IEEE Transactions on Power Electronics, vol. 35, no. 7, pp. 7434–7450, 2020.
- It would be enough to present the time function of the state variables on Fig 13-17 between 0 and 0.05s. It would be nice to add a figure which shows only few periods of the current in steady-state to see its harmonic performance.
- in equation 7: omega_r is used in the equation, which is called as rotor angular frequency, which is most likely the mechanical speed of the machine (!?). I think in the equation 7 the electrical angular frequency should be used, which is P times the mechanical angular frequency. Furthermore, I suggest to use clear symbols: rotor angular frequency is denoted as wm on Fig 14, in the block diagram (Fig.10) it is denoted as omega, in the equation as omega_r. I suggest to use the most common symbols and denoted electrical angular frequency as omega_1 and the mechanical speed as capital letter omega.
- Figure 13-18 gives the waveforms of the motor operation at 3000 rpm, but in the case of the speed, the unit is given in rad/m. Rad/m is not the same as rpm, please correct it!
- page 12, line 312: double comma after the word "work"
- the loss judgement is carried out at frequency 200 Hz (0.005s), based on your answer:"Therefore, for the time of loss judgment, this paper tested a number of times in the simulation verification process, and finally took 0.005 as the loss judgment time". Most likely this sampling rate gives proper value as the electrical frequency is also 200 Hz (3000 rpm -> 3000/60*4 = 200 Hz), so the losses evaluated for one electrical period. This selected 200 Hz sampling rate provide correct values at a lower or higher angular speeds as well? Perhaps the evaluation of the loss judgement should be somehow synchronized to the electrical angular frequency.
- page 7,line 190: "switching switching process"
- The calculation of the loss is based on equation 4 and 5. These equations are quite general. Please add some more comments how you calculate the losses in "real-time" for the control algorithm (e.g. the fs switching frequency is not constant).
Author Response
Dear Reviewers:
Thank you very much for your professional comments on our article. As you may have noticed, there are several issues that need to be addressed. Based on your good suggestions, we have made extensive changes to our previous draft. As a result, we have uploaded a copy of the original draft with all changes highlighted using the Track Changes mode in MS Word. Our point-by-point response to the comments is attached below. While reproducing the comments, our responses are directly highlighted in a different color (red).
1.on page five line 142: "The inverter switching frequency is set to 100kHz" 100 Khz is the sampling frequency of the MPC control algorithm, but as Finite-Set MPC is utilized, most likely the switching frequency is much less than the sampling frequency. It can happen this results were made by a constant switching frequency (it is not clear from the text) without the proposed MPC scheme. In that case, the selected 100 kHz is very large for the IGBT taking into account for example the delays in the switching.
page 15, line 366: "switching frequency of 100khz". As I indicated in my previous comment, 100 kHz is the sampling frequency of the control algorithm, and the switching frequency should be much lower. Please calculate the average number of switching and indicate in the paper. I suggest to use the method introduced in the paper (see equation 5):P. Karamanakos and T. Geyer, “Guidelines for the design of finite control set model predictive controllers,” IEEE Transactions on Power Electronics, vol. 35, no. 7, pp. 7434–7450, 2020.
Thank you for your comments, we have overlooked this point thank you for pointing it out promptly. According to the calculations we obtained the average switching frequency for a load of 5Nm and a target speed of 3000 rad/m. The calculated value is 18.7 kHz. because of the same setup conditions, the result is placed at the beginning of the analyzed results in Fig. 13-18.
2.It would be enough to present the time function of the state variables on Fig 13-17 between 0 and 0.05s. It would be nice to add a figure which shows only few periods of the current in steady-state to see its harmonic performance.
Thanks to your comments, we have added the graph after the steady state current approach to Figure 16.At the same time, most of the subsequent simulations in this paper show a time of 0-0.15s, in order to keep all the time and produce a contrast, so the choice of 0.15s.
3.in equation 7: omega_r is used in the equation, which is called as rotor angular frequency, which is most likely the mechanical speed of the machine (!?). I think in the equation 7 the electrical angular frequency should be used, which is P times the mechanical angular frequency. Furthermore, I suggest to use clear symbols: rotor angular frequency is denoted as wm on Fig 14, in the block diagram (Fig.10) it is denoted as omega, in the equation as omega_r. I suggest to use the most common symbols and denoted electrical angular frequency as omega_1 and the mechanical speed as capital letter omega.
Thank you for your keen discovery of this problem, we have made changes according to your request, will be represented as omega in Fig. 10, will be represented as omega_1 in Equation 7, at the same time, the problem of the subsequent simulation diagrams will be corrected, in order to prevent the recognition of error will be followed by the diagrams of the mechanical speed is not expressed, only expressed as "rotor speed".
4.Figure 13-18 gives the waveforms of the motor operation at 3000 rpm, but in the case of the speed, the unit is given in rad/m. Rad/m is not the same as rpm, please correct it!
Thank you for your comments, we have re-modified the relevant content in the text, and the modifications are marked in red.
5.page 12, line 312: double comma after the word "work"
Thank you for your comments, we have re-modified the relevant content in the text, and the modifications are marked in red.
6.the loss judgement is carried out at frequency 200 Hz (0.005s), based on your answer:"Therefore, for the time of loss judgment, this paper tested a number of times in the simulation verification process, and finally took 0.005 as the loss judgment time". Most likely this sampling rate gives proper value as the electrical frequency is also 200 Hz (3000 rpm -> 3000/60*4 = 200 Hz), so the losses evaluated for one electrical period. This selected 200 Hz sampling rate provide correct values at a lower or higher angular speeds as well? Perhaps the evaluation of the loss judgement should be somehow synchronized to the electrical angular frequency.
Thank you for your comments, we have discussed this idea and found it very valuable, so it will be used as a basis in section4.4. Please forgive us for the time constraints of our study, which prevented us from going further, so we will add this idea to the summary in the form of a forward-looking statement for others to consider.
7.page 7,line 190: "switching switching process"
Thank you for your comments, we have re-modified the relevant content in the text, and the modifications are marked in red.
8.The calculation of the loss is based on equation 4 and 5. These equations are quite general. Please add some more comments how you calculate the losses in "real-time" for the control algorithm (e.g. the fs switching frequency is not constant).
Thank you for your comments, we have added clarification with the method of calculating the loss for the fact that it is calculated for each sampling time.
Finally, thank you for your comments once again, they have brought a great boost to my post!
Sincerely yours,
Shaoqi Wan
Round 3
Reviewer 2 Report
Comments and Suggestions for Authors
Authors improved their paper based on my comments and answered to my queries.
I still have a few comments:
- In previous review round I suggest the following: "Figure 13-18 gives the waveforms of the motor operation at 3000 rpm, but in the case of the speed, the unit is given in rad/m. Rad/m is not the same as rpm, please correct it"
As a response, the authors replaced rpm to rad/m in the main text. However, it is not correct. In the investigated steady state working point the synchronous frequency is 200 Hz (see Fig.18). As the machine has 4 pole pairs, the mechanical rotational speed in RPM (revolution per minute) should be 200/4*60 = 3000 rpm. The same was written in the previous version, which was correct. Now, the authors replaced 3000 rpm to 3000 rad/m everywhere. 3000 rad/m = 3000/(2pi) rpm = 477.5 rpm, which results in 31.83 Hz synchronous frequency... My original suggestion was to rewrite the rad/m on the figures to rpm!
- In Fig.4, 6 and 8. the authors presents the losses of the transistors to demonstrate the effect of the different modes. It is not clear from the text, whether this simulation is obtained in open-loop or closed-loop, using the proposed MPC scheme or not. According to the text in this case the switching frequency set to 100 kHz. The total switching loss in Fig.4 is 50.8W, in Fig.6 63.4 W and on Fig.8 49.3 W. For the proposed MPC scheme, the sampling frequency is 100 kHz, and the resulting average switching frequency is 18.7 kHz. For the MPC operation the total switching loss in Fig.11 is 56,7 W and in Fig.12 it is 51.8W. First of all, the improvement is clear comparing the losses on Fig.12 and Fig.11. At the same time, what is contradictory is that the total switching loss is almost the same for a switching frequency of 18.7 kHz and a switching frequency of 100 kHz, even though the switching loss is directly proportional to the number of switches, which in the case of Fig.11 and 12 is one sixth to the case presented on Fig.4,6 and 8.
- Please only highlight the changes in the manuscript comparing to the previous version and not to the original version!
Author Response
Dear Reviewers:
Thank you very much for your professional comments on our article. As you may have noticed, there are several issues that need to be addressed. Based on your good suggestions, we have made extensive changes to our previous draft. As a result, we have uploaded a copy of the original draft with all changes highlighted using the Track Changes mode in MS Word. Our point-by-point response to the comments is attached below. While reproducing the comments, our responses are directly highlighted in a different color (red).
1.In previous review round I suggest the following: "Figure 13-18 gives the waveforms of the motor operation at 3000 rpm, but in the case of the speed, the unit is given in rad/m. Rad/m is not the same as rpm, please correct it"
As a response, the authors replaced rpm to rad/m in the main text. However, it is not correct. In the investigated steady state working point the synchronous frequency is 200 Hz (see Fig.18). As the machine has 4 pole pairs, the mechanical rotational speed in RPM (revolution per minute) should be 200/4*60 = 3000 rpm. The same was written in the previous version, which was correct. Now, the authors replaced 3000 rpm to 3000 rad/m everywhere. 3000 rad/m = 3000/(2pi) rpm = 477.5 rpm, which results in 31.83 Hz synchronous frequency... My original suggestion was to rewrite the rad/m on the figures to rpm!
Thank you for your comments, it was due to our negligence that this error occurred, we feel very bad and offer you a sincere apology!
2.In Fig.4, 6 and 8. the authors presents the losses of the transistors to demonstrate the effect of the different modes. It is not clear from the text, whether this simulation is obtained in open-loop or closed-loop, using the proposed MPC scheme or not.
Thanks to your comments, we have added to the loss background based on the modifications made in the previous manuscript, i.e., it was performed under model predictive control with the rotational speed set to 3000 rpm.
3.According to the text in this case the switching frequency set to 100 kHz. The total switching loss in Fig.4 is 50.8W, in Fig.6 63.4 W and on Fig.8 49.3 W. For the proposed MPC scheme, the sampling frequency is 100 kHz, and the resulting average switching frequency is 18.7 kHz. For the MPC operation the total switching loss in Fig.11 is 56,7 W and in Fig.12 it is 51.8W. First of all, the improvement is clear comparing the losses on Fig.12 and Fig.11. At the same time, what is contradictory is that the total switching loss is almost the same for a switching frequency of 18.7 kHz and a switching frequency of 100 kHz, even though the switching loss is directly proportional to the number of switches, which in the case of Fig.11 and 12 is one sixth to the case presented on Fig.4,6 and 8.
Thank you for your comments, the simulation backgrounds in Fig. 4, Fig. 6, Fig. 8 and Fig. 11 and Fig. 12 are the same, the difference is only in the control of the ANPC zero level path.
I understand what you said that the switching loss is proportional to the number of switches, but after you proposed that we calculate the average switching frequency of the three ways of mixing, we can take into account that for Fig. 4, Fig. 6, and Fig. 8, the average switching frequency is not up to 100khz, but is close to 18.7kHz, and since the focus of our study is the switching loss, so we do not increase the corresponding average switching frequency of Fig. 4, Fig. 6, and Fig. 8 at the moment, and therefore we will consider the following study. We also change the setup condition to a sampling frequency of 100kHz in the context of testing the switching losses, which is also the same as the frequency at which the losses are captured in the later simulations.
Meanwhile, we also consider to reduce the sampling frequency to improve the efficiency of the simulation operation in the subsequent study (because the model proposed in this paper cannot reach 100kHz under predictive control).
Finally, thank you for your comments once again, they have brought a great boost to my post!
Sincerely yours,
Shaoqi Wan